# Association between composite dietary antioxidant indices and anemia: NHANES 2003–2018

Qian Wu[1,2], Zhiyu Wang[2], Jieyu Xia[2], Hui Xu[1], Gang Huang[1], Guangyong Feng[1], Xiaoxia Gou[1] *

1 Department of Head and Neck Oncology, The Second Affiliated Hospital of Zunyi Medical University, Zunyi, Guizhou, China, 2 Department of Oncology, The Fifth Affiliated Hospital of Zunyi Medical University, Zhuhai, Guangdong, China

* gouxx2020@163.com

## Abstract

### Background

There is increasing acknowledgment of the potential role that diet rich in antioxidants may play in the prevention of anemia. As a significant indicator of antioxidant-rich diet, the relationship between the composite dietary antioxidant index (CDAI) and anemia has not been extensively studied. Therefore, this study aims to explore the association between CDAI and anemia.

### Methods

Utilizing datas from the 2003–2018 National Health and Nutrition Examination Survey (NHANES) database. The CDAI was calculated using six dietary antioxidants, based on two 24-hour dietary recall interviews, serving as comprehensive measure of the intake of these antioxidants. Weighted multivariable logistic regression and restricted cubic spline (RCS) analysis was conducted to investigate the association between CDAI and anemia. Furthermore, subgroup analyses were performed to enhance datas reliability.

### Results

A total of 33914 participants were included in the study, among which 3,416 (10.07%) were diagnosed with anemia. The unadjusted model showed negative association between CDAI and anemia (odds ratio [OR]: 0.94; 95% confidence interval [95%CI]: 0.93–0.96; $P < 0.001$). After adjusting for all covariates, with each increase in CDAI level linked to 3% lower risk of anemia (OR: 0.97; 95%CI: 0.95–0.98; $P < 0.001$). Moreover, when CDAI was categorized into quartiles, the observed trend persisted ($P < 0.001$). The RCS analysis revealed linear negative relationship between CDAI and anemia ($P$ for nonlinearity = 0.619). Except for sex, smoking, diabetes and hypertension, no statistically significant interactions were found in any subgroup analysis ($P < 0.05$ for interaction).

**Data Availability Statement:** Data availability: The National Health and Nutrition Examination Survey (NHANES) database, which is managed by the National Center for Health Statistics (NCHS) of the Centers for Disease Control and Prevention,

provided the data used in this study. The authors confirm they did not have any special access privileges that others would not have and others would be able to access these data in the same manner as the authors. The data relevant to this study are publicly available and can be accessed at https://wwwn.cdc.gov/Nchs/Nhanes/2011-2012/CBC_G.htm#LBXHGB. All relevant data from this database have been provided within the paper and its Supporting Information files.

**Funding:** This study was supported by the the National Natural Science Foundation of China (82460465); The Natural Science Foundation of Guizhou Province (Qian Ke He Basic Proiect ZK [2024]346); The Guizhou Anti-Cancer Association Science and Technology Plan Project (Anti-Cancer Association Science and Technology 006[2023]).

**Competing interests:** The authors have declared that no competing interests exist.

## Conclusion

Our findings suggest that CDAI levels are inversely related to the prevalence of anemia. Consequently, monitoring individuals with low CDAI scores may facilitate the timely identification of anemia and enhance clinical decision-making.

## Introduction

Anemia is a widespread global health concern that has been consistently linked to various negative outcomes in recent years [1]. It is characterized by an imbalance between the production and destruction of red blood cells (RBCs), resulting in inadequate oxygen delivery to vital tissues like the brain and heart [2]. The diagnosis of anemia typically relies on the hemoglobin level in the blood [3]. The prevalence and incidence of anemia have risen significantly, attributed to combination of increased nutrient deficiencies, chronic diseases, inherited hemoglobin disorders, and the use of specific medications [4, 5]. Anemia can have detrimental effects on cognitive and physical functions [6], leading to decreased economic productivity and higher morbidity and mortality rates [7], presenting significant health challenge in modern society. Early recognition of anemia presents an opportunity to delay or prevent the onset of the disease and enhance treatment outcomes. Consequently, identifying new indicators closely associated with anemia is of great significance for developing more effective anemia prevention strategies [4].

Oxidative stress is defined as an imbalance between the production of reactive oxygen species (ROS) and antioxidant defense mechanisms, and is acknowledged as a major factor in various conditions like inflammation, aging, cancer, and cardiovascular diseases [8]. Numerous studies have emphasized a significant link between oxidative stress and the onset of anemia [9–11]. A modifiable risk factor for reducing oxidative stress is diet [12]. By including dietary antioxidants, individuals can alleviate or prevent oxidative stress-related diseases by neutralizing the harmful effects of ROS [13]. Consistent consumption of antioxidants can reduce oxidative stress levels and enhance the body's ability to withstand it [14]. Halima et al found that antioxidants exert protective effect against anemia and provide significant alternative benefits for RBC function. This is achieved by preventing lipid peroxidation in RBCs, increasing levels of reduced glutathione (GSH), and reducing RBC permeability [15]. As research enhances our understanding of nutrition and oxidative stress, there is an increasing interest in the role of antioxidant-rich diets in the prevention of anemia.

The composite dietary antioxidant index (CDAI) was developed by Wright et al as a tool to assess antioxidant intake, serving as a composite score to evaluate dietary antioxidant consumption [16]. This index considers key nutrients such as vitamins A, vitamins C, and vitamins E, zinc, selenium, and carotenoids. While previous studies have mainly examined the relationship between CDAI and different diseases. Wang et al posited that the CDAI was positively correlated with lower prevalence of chronic kidney disease among American adults [17]. Similarly, Yu et al discovered that elevated CDAI scores were linked to decreased risk of colorectal cancer (CRC) and concluded that food-based antioxidants might contribute to lowering the risk of CRC in the general population [18]. Additionally, another study indicated that higher intake of dietary antioxidants, assessed through the Dietary Antioxidant Quality Score (DAQS) and CDAI, was associated with reduced risk of all-cause and cardiovascular disease mortality in adults with diabetes [19]. However, the specific connection between CDAI and anemia is yet to be conclusively determined.

A cross-sectional study was conducted using data from the National Health and Nutrition Examination Survey (NHANES) to assess the relationship between CDAI and anemia. The hypothesis posited that higher CDAI scores would correlate with decreased prevalence of anemia.

## Materials and methods

### NHANES database

The analysis is based on data from the NHANES, a cross-sectional survey conducted by the Centers for Disease Control (CDC) and the National Center for Health Statistics (NCHS). The survey's objective is to evaluate the health and nutrition status of a representative population, encompassing institutionalized individuals [20]. Since 1999, NHANES has been an ongoing study that releases data every two years [21]. The database is available for free download from the official website. Approval for the study was obtained from the NCHS Research Ethics Review Board, and all participants provided written informed consent, and no external ethic approval was required for this study.

### Study population

This study utilized NHANES data spanning from 2003 to 2018, encompassing a total of 80,312 participants. After excluding individuals without hemoglobin data (N = 14,320), those lacking CDAI data (N = 4,656), and pregnant participants (N = 1,051), we further removed cases with missing covariate data. Ultimately, we obtained final sample size of 33914 participants. A flow chart depicting the exclusion criteria is presented in Fig 1.

### Exposure and outcomes

According to World Health Organization (WHO) guidelines, patients with anemia are defined as having hemoglobin (HB) levels below 12 g/dL for women and below 13 g/dL for men [22].

Data on dietary antioxidant intake were obtained from the average of two 24-hour dietary recall interviews conducted as part of the NHANES. The initial dietary recall was performed at mobile examination center (MEC) by trained interviewers who adhered to standardized protocol. During this face-to-face interview, detailed information regarding all food and beverages consumed by participants over the past 24 hours was collected. This was followed by second interview conducted via telephone 3 to 10 days later [23]. Utilizing average dietary intake data from two non-consecutive days is deemed more accurate than relying solely on data from a single day [24]. To evaluate the overall exposure to dietary antioxidants, the intake of six key antioxidants (vitamin A, vitamin C, vitamin E, zinc, selenium, and carotene) was analyzed and quantified using to calculate the CDAI [25]. This involved standardizing the intake levels of the six antioxidants by subtracting the mean and dividing by the standard deviation, followed by summing up the standardized values:

$$\text{CDAI} = \sum_{i=1}^{n} \left( \frac{x_n - \mu_n}{s_n} \right)$$

### Covariates

Potential confounders considered in this study including age, sex, race, education levels, body mass index (BMI), poverty income ratio (PIR), smoking consumption, alcohol consumption, as well as comorbidities like hypertension, diabetes, cancer, and hyperlipidemia. Age was

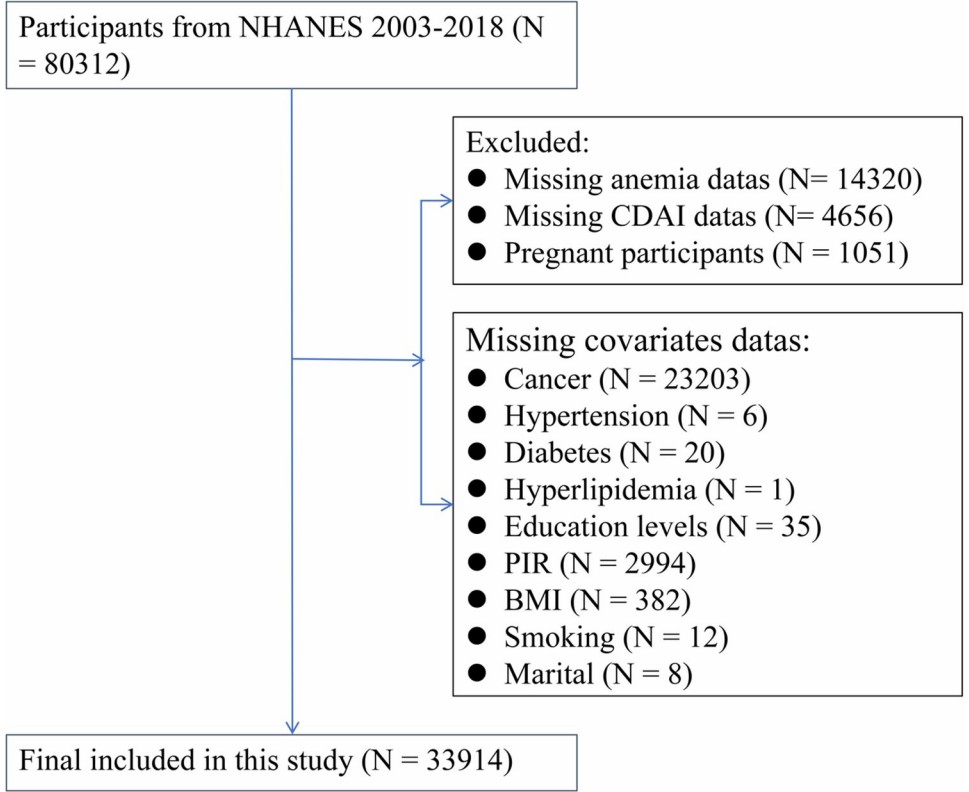

**Fig 1. Flow chart of participant selection in NHANES.** Abbreviations: NHANES, National Health and Nutrition Examination Survey; CDAI, Composite Dietary Antioxidant Index.

categorized into 40 years, ethnicity into non-Hispanic White, non-Hispanic Black, Mexican American, and others, and education levels into less than high school, high school, and more than high school. Marital status was divided into married/living with partner or widowed/divorced/separated/never married. PIR was categorized as low ($\leqq$ 2.14) and high ($\geqq$ 2.14). BMI was categorized as thin/normal ($\leqq$ 18.5kg/m$^2$, 18.6–24.9kg/m$^2$), overweight (25.0–29.9 kg/m$^2$), and obese ($\geqq$ 30 kg/m$^2$).

Smoking status was categorized into two groups: smokers, defined as those who had smoked at least 100 cigarettes in their lifetime (coded as 'yes'), and non-smokers (coded as 'no') [26]. Alcohol consumption was classified as 'yes' for those who had consumed at least 12 alcoholic drinks in the past year, and 'no' for others [27]. Participants were considered to have a history of diabetes, hypertension, and hyperlipidemia if they reported physicians diagnosis of these conditions. The diagnosis of cancer required meeting two criteria: (1) positive response to 'ever been told they have cancer or a malignancy of any kind' (variable mcq220); (2) providing details for the inquiry 'what kind of cancer?' (variable mcq230A).

## Statistical analysis

Participants were categorized into four groups based on quartiles CDAI. Continuous variables were presented as mean ± standard deviation (mean ± SD), while categorical variables were expressed as percentages. The associations between baseline characteristics and CDAI quartiles were evaluated using chi-square or t tests. Weighted logistic regression model was employed to investigate the relationship between CDAI and anemia, with results reported as adjusted

odds ratio (OR) and 95% confidence intervals (CI). Model 1 had no adjustments, while model 2 included adjustments for gender, age, education level, and race. Model 3 further adjusted for PIR, BMI, smoking, drinking, hypertension, diabetes, and hyperlipidemia as covariates in addition to those in Model 2. CDAI was analyzed both as continuous and categorical variable to explore correlations. All statistical analyses were conducted using R 4.3.3 with appropriate weights, and statistical significance was defined as $P < 0.05$.

## Ethics statement

The survey procedures are in accordance with the standards outlined in the declaration of Helsinki [28]. All information from the NHANES program is freely available to the public, therefore, the approval of the medical ethics committee board was not required [29].

## Results

### Baseline characteristics

This study involved 33914 patients who met strict inclusion and exclusion criteria, with 3416 (10.07%) of them diagnosed anemia. The average age of the participants was 47.35 ± 0.22 years. Patients were categorized into 4 groups based on CDAI quartiles, and their baseline characteristics are presented in Table 1. Higher GDAI levels were associated with certain characteristics such as younger age, non-Hispanic white married female, non-drinkers, non-smokers, higher BMI, higher PIR, higher education levels, absence of diabetes, cancer, hypertension, and anemia, but presence of hyperlipidemia. Univariable and multivariable logistic analyses showed significant link between CDAI and anemia ($P < 0.001$) (S1 Table).

### The association between CDAI and anemia

Table 2 presents the results of the weighted logistic regression analysis investigating the relationship between anemia and CDAI. The analysis demonstrated robust negative correlation between CDAI and anemia when considering CDAI as continuous variable in model 1 (OR: 0.94; 95%CI: 0.93–0.96), model 2 (OR: 0.96; 95%CI: 0.95–0.97) and model 3 (OR: 0.97; 95%CI: 0.95–0.98). Particularly noteworthy is the significant trend in model 3 ($P$ for trend < 0.001) when CDAI was categorized into quartiles, indicating that higher CDAI scores were linked to lower likelihood of anemia. To further confirm this association, restricted cubic spline (RCS) analysis with 3 knots was performed, revealing linear negative correlation between CDAI and anemia ($P$ for nonlinearity = 0.619), as depicted in Fig 2.

### The association between antioxidant components and anemia

The relationship between the six antioxidant components of CDAI and anemia was further examined (Table 3), it was observed that vitamin A was linked to anemia in model 1 (OR: 1.00; 95%CI: 1.00–1.00, $P = 0.010$). Vitamin C showed an association with anemia in models 1 and 2 (OR: 1.00; 95%CI: 1.00–1.00, $P = 0.030$ or $P = 0.002$). After adjusting for all variables, zinc (OR: 0.97; 95% CI: 0.97–0.98, $P < 0.001$), vitamin E (OR: 0.98; 95%CI: 0.98–0.99, $P < 0.001$), carotene (OR: 1.00; 95% CI:1.00–1.00, $P < 0.001$), and selenium (OR: 1.00; 95% CI:1.00–1.00, $P = 0.010$) were identified as independent factors associated with anemia.

### Subgroup analysis

To further investigate the influence of CDAI on anemia risk within specific subgroups, stratified analysis was performed based on gender, age, race, PIR, BMI, hypertension, diabetes, and hyperlipidemia. Results indicated that except for sex, smoking consumption, and diabetes,

**Table 1. The weighted baseline characteristics by CDAI quartiles.**

| Characteristics | Total (N = 33914) | Quartiles of CDAI | | | | P value |
|---|---|---|---|---|---|---|
| | | Q1 (N = 8482) | Q2 (N = 8477) | Q3 (N = 8477) | Q4 (N = 8478) | |
| **Age (years)** | 47.35 ± 0.22 | 47.35 ± 0.22 | 47.84 ± 0.30 | 47.99 ± 0.26 | 47.66 ± 0.32 | < **0.001** |
| **Sex** | | | | | | < **0.001** |
| Female | 17110 (50.45) | 4622 (54.49) | 4182 (49.33) | 4175 (49.25) | 4131 (48.73) | |
| Male | 16804 (49.55) | 3860 (45.51) | 4295 (50.67) | 4302 (50.75) | 4347 (51.27) | |
| **Race** | | | | | | < **0.001** |
| Non-Hispanic black | 6977 (20.57) | 2124 (25.04) | 1712 (20.20) | 1546 (18.24) | 1595 (18.81) | |
| Non-Hispanic white | 15510 (45.73) | 3555 (41.91) | 3921 (46.25) | 4070 (48.01) | 3964 (46.76) | |
| Mexican American | 5323 (15.70) | 1287 (15.17) | 1368 (16.14) | 1352 (15.95) | 1316 (15.52) | |
| Other Hispanic | 6104 (18.00) | 2124 (25.04) | 1476 (17.41) | 1509 (17.80) | 1603 (18.91) | |
| **Education levels** | | | | | | < **0.001** |
| High school | 4704 (13.87) | 1471 (17.34) | 1207 (14.24) | 1067 (12.59) | 959 (11.31) | |
| Less than high school | 3467 (10.22) | 1221 (14.40) | 909 (10.72) | 748 (8.82) | 589 (6.95) | |
| More than high school | 25743 (75.91) | 5790 (68.26) | 6361 (75.04) | 6662 (78.59) | 6930 (81.74) | |
| **Marital status** | | | | | | < **0.001** |
| Divorced | 3763 (11.10) | 1098 (12.95) | 899 (10.61) | 889 (10.61) | 877 (10.34) | |
| Living with partner | 2660 (7.84) | 677 (7.98) | 611 (7.21) | 642 (7.57) | 730 (8.61) | |
| Married | 17708 (52.21) | 3971 (46.82) | 4541 (53.57) | 4653 (54.89) | 4543 (53.59) | |
| Never married | 5905 (17.42) | 1549 (18.26) | 1416 (16.70) | 1378 (16.26) | 1562 (18.42) | |
| Separated/Windowed | 3878 (11.43) | 1187 (13.99) | 1010 (11.91) | 915 (10.79) | 766 (9.04) | |
| **Drinking consumption** | | | | | | < **0.001** |
| No | 24345 (71.78) | 6494 (72.88) | 6184 (72.95) | 5874 (69.29) | 5793 (68.37) | |
| Yes | 9569 (28.22) | 1988 (27.12) | 2293 (27.05) | 2603 (30.71) | 2685 (33.83) | |
| **Smoking consumption** | | | | | | < **0.001** |
| No | 18318 (54.01) | 4204 (49.56) | 4537 (53.52) | 4716 (55.63) | 4861 (57.34) | |
| Yes | 15596 (45.99) | 4278 (50.44) | 3940 (46.48) | 3761 (44.37) | 3617 (42.66) | |
| **BMI** | | | | | | < **0.001** |
| Normal | 9255 (27.29) | 2249 (26.51) | 2218 (26.17) | 2263 (26.70) | 2525 (29.78) | |
| Thin | 518 (1.53) | 156 (1.84) | 110 (1.30) | 113 (1.33) | 139 (1.64) | |
| Obese | 12865 (37.93) | 3312 (39.05) | 3247 (38.30) | 3229 (38.09) | 3077 (36.30) | |
| Overweight | 11276 (33.25) | 2765 (32.60) | 2902 (34.23) | 2872 (33.88) | 2737 (32.28) | |
| **PIR** | | | | | | < **0.001** |
| ≥2.14 | 16988 (50.09) | 3426 (40.39) | 4192 (49.45) | 4643 (54.77) | 4727 (55.76) | |
| <2.14 | 16926 (49.91) | 5056 (59.61) | 4285 (50.55) | 3834 (45.23) | 3751 (44.24) | |
| **Hypertension** | | | | | | < **0.001** |
| No | 19345 (57.04) | 4463 (52.62) | 4711 (55.57) | 4972 (58.65) | 5199 (61.32) | |
| Yes | 14569 (42.96) | 4019 (47.38) | 3766 (44.43) | 3505 (41.35) | 3279 (38.68) | |
| **Hyperlipidemia** | | | | | | < **0.001** |
| No | 9725 (28.68) | 2237 (26.37) | 2331 (27.50) | 2443 (28.82) | 2714 (32.01) | |
| Yes | 24189 (71.32) | 6245 (73.62) | 6146 (72.50) | 6034 (71.18) | 5764 (67.99) | |
| **Diabetes** | | | | | | < **0.001** |
| Borderline | 746 (2.20) | 181 (2.13) | 192 (2.26) | 179 (2.11) | 194 (2.29) | |
| No | 28883 (85.17) | 7051 (83.13) | 7112 (83.90) | 7306 (86.19) | 7414 (87.45) | |
| Yes | 4285 (12.63) | 1250 (14.74) | 1173 (13.84) | 992 (11.70) | 870 (10.26) | |
| **Anemia** | | | | | | < **0.001** |
| No | 30498 (89.93) | 7440 (87.72) | 7589 (89.52) | 7655 (90.30) | 7814 (92.17) | |
| Yes | 3416 (10.07) | 1042 (12.28) | 888 (10.48) | 822 (9.70) | 664 (7.83) | |

(*Continued*)

**Table 1.** (Continued)

| Characteristics | Total (N = 33914) | Quartiles of CDAI | | | | P value |
|---|---|---|---|---|---|---|
| | | Q1 (N = 8482) | Q2 (N = 8477) | Q3 (N = 8477) | Q4 (N = 8478) | |
| **Cancer** | | | | | | **0.030** |
| No | 30678 (90.46) | 7681 (90.56) | 7590 (89.54) | 7662 (90.39) | 7745 (91.35) | |
| Yes | 3236 (9.54) | 801 (9.44) | 887 (10.46) | 815 (9.61) | 733 (8.65) | |

The first quartile of CDAI is defined as Q1, with the subsequent quartiles being defined as Q2, Q3, and Q4. Note: Data are presented as mean (SD) or n (%). BMI, body mass index; HB, hemoglobin; PIR, poverty income ratio; CDAI, composite dietary antioxidant indexes. *P* values < 0.05 are in bold.

there were no significant interactions between CDAI and anemia risk (*P* for interaction > 0.05, Fig 3). This indicates that the protective effect of CDAI on anemia is particularly pronounced in male non-smokers and non-diabetic individuals.

## Discussion

This study is the first to investigate the relationship between CDAI and the prevalence of anemia based on data from NHANES. Upon adjusting for potential confounders, negative association between CDAI and anemia in American adults was observed, with linear trend established through dose-response analysis. Additionally, subgroup analysis indicated the protective effect of CDAI against anemia was particularly significant among male nonsmokers and nondiabetic individuals. Clinically, these results suggest that anemia can be effectively prevented, diagnosed, and treated through appropriate dietary modifications rich in antioxidants.

Diet plays a crucial role in managing the body's oxidative stress levels [30], with dietary antioxidants being essential in lowering the risk of aging, cancer, diabetes, inflammation, liver disease, and cardiovascular disease. The fast-paced lifestyle of modern society leads to an increase in free radicals in the body, which can demage cells, tissues, and organs, ultimately impacting longevity. Antioxidants are key in fighting free radicals, thereby contributing to prevent both short-term and long-term diseases [31]. In this study, we focus on the CDAI for two primary reasons. Firstly, CDAI serves as a crucial indicator of dietary antioxidant capacity. Unlike single dietary antioxidant indicators, CDAI encompasses key antioxidant nutrients, including vitamin A, vitamin C, vitamin E, zinc, selenium, and carotene, thereby facilitating more comprehensive evaluation of overall diet quality. Secondly, research in related fields has

**Table 2. The association between CDAI and anemia.**

| Exposure | Model 1 | | Model 2 | | Model 3 | |
|---|---|---|---|---|---|---|
| | OR [95%CI] | P | OR [95%CI] | P | OR [95%CI] | P |
| CDAI | 0.94 (0.93,0.96) | **<0.001** | 0.96 (0.95,0.97) | **<0.001** | 0.97 (0.95,0.98) | **<0.001** |
| CDAIQ | | | | | | |
| Q1 | ref | | ref | | ref | |
| Q2 | 0.85 (0.75,0.97) | **0.010** | 0.98 (0.86,1.11) | 0.690 | 0.99 (0.86,1.13) | 0.870 |
| Q3 | 0.74 (0.65,0.84) | **<0.001** | 0.89 (0.78,1.02) | 0.140 | 0.93 (0.81,1.07) | 0.490 |
| Q4 | 0.56 (0.49,0.64) | **<0.001** | 0.68 (0.59,0.77) | **<0.001** | 0.71 (0.62,0.81) | **<0.001** |
| *P* for trend | | **<0.001** | | **<0.001** | | **<0.001** |

Model 1: no covariates were adjusted.

Model 2: adjusted for gender, age, education level and race.

Model 3: adjusted for gender, age, race, education level, PIR, BMI, smoke, drink, hypertension, diabetes and hyperlipidemia. *P* values < 0.05 are in bold.

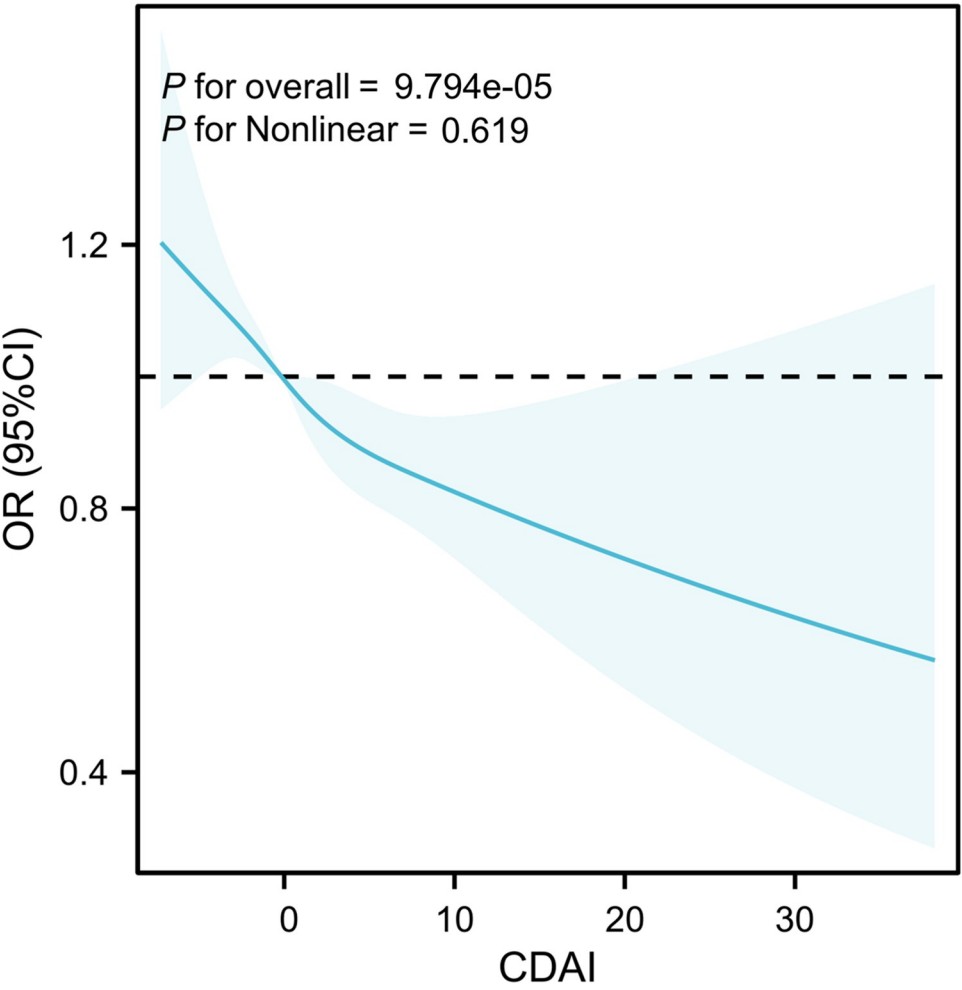

**Fig 2. The RCS curve of the association between CDAI and anemia.** RCS regression was adjusted by model 3. RCS, restricted cubic spline; CDAI, composite dietary antioxidant index; OR, odds ratio.

garnered significant attention. A study within NHANES demonstrated negative linear correlation between CDAI and hypertension [32]. These results also propose that increasing CDAI levels through a diet rich in antioxidant nutrients could potentially reduce the occurrence of

**Table 3. The weighted logistic regression analysis of the association between antioxidant components and anemia.**

| Antioxidant components | Model 1 | | Model 2 | | Model 3 | |
|---|---|---|---|---|---|---|
| | OR [95%CI] | *P* | OR [95%CI] | *P* | OR [95%CI] | *P* |
| Vitamins A (mg) | 1.00 (1.00,1.00) | **0.010** | 1.00 (1.00,1.00) | 0.890 | 1.00 (1.00,1.00) | 0.910 |
| Vitamins C (mg) | 1.00 (1.00,1.00) | **0.030** | 1.00 (1.00,1.00) | **0.020** | 1.00 (1.00,1.00) | 0.050 |
| Vitamins E (mg) | 0.96 (0.95,0.97) | **<0.001** | 0.98 (0.97,0.99) | **< 0.001** | 0.98 (0.98,0.99) | **<0.001** |
| Zinc (mg) | 0.94 (0.94,0.95) | **<0.001** | 0.97 (0.96,0.98) | **<0.001** | 0.97 (0.97,0.98) | **<0.001** |
| Selenium (mg) | 0.99 (0.99,1.00) | **<0.001** | 1.00 (1.00,1.00) | **<0.001** | 1.00 (1.00,1.00) | **<0.001** |
| Carotene (mg) | 1.00 (1.00,1.00) | **<0.001** | 1.00 (1.00,1.00) | **0.001** | 1.00 (1.00,1.00) | **0.010** |

Model 1: no covariates were adjusted.

Model 2: adjusted for gender, age, education level and race.

Model 3: adjusted for gender, age, race, education level, PIR, BMI, smoke, drink, hypertension, diabetes and hyperlipidemia. *P* values < 0.05 are in bold.

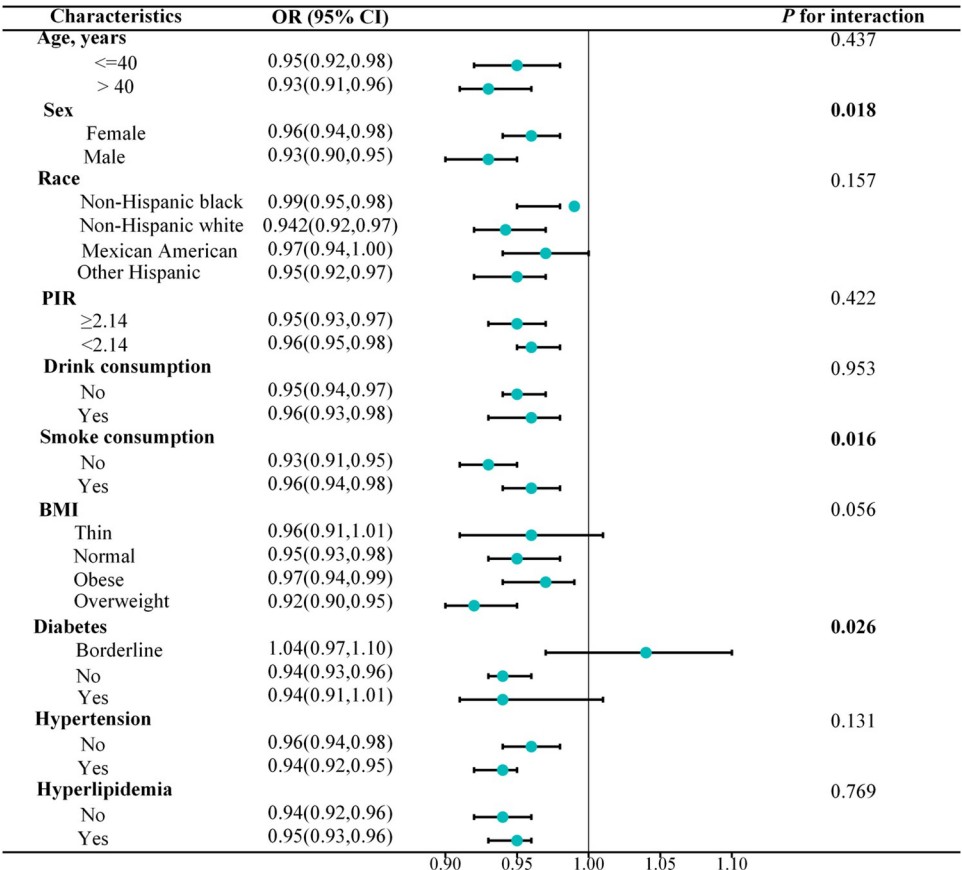

| Characteristics | OR (95% CI) | | P for interaction |
|---|---|---|---|
| **Age, years** | | | 0.437 |
| <=40 | 0.95(0.92,0.98) | | |
| > 40 | 0.93(0.91,0.96) | | |
| **Sex** | | | **0.018** |
| Female | 0.96(0.94,0.98) | | |
| Male | 0.93(0.90,0.95) | | |
| **Race** | | | 0.157 |
| Non-Hispanic black | 0.99(0.95,0.98) | | |
| Non-Hispanic white | 0.942(0.92,0.97) | | |
| Mexican American | 0.97(0.94,1.00) | | |
| Other Hispanic | 0.95(0.92,0.97) | | |
| **PIR** | | | 0.422 |
| ≥2.14 | 0.95(0.93,0.97) | | |
| <2.14 | 0.96(0.95,0.98) | | |
| **Drink consumption** | | | 0.953 |
| No | 0.95(0.94,0.97) | | |
| Yes | 0.96(0.93,0.98) | | |
| **Smoke consumption** | | | **0.016** |
| No | 0.93(0.91,0.95) | | |
| Yes | 0.96(0.94,0.98) | | |
| **BMI** | | | 0.056 |
| Thin | 0.96(0.91,1.01) | | |
| Normal | 0.95(0.93,0.98) | | |
| Obese | 0.97(0.94,0.99) | | |
| Overweight | 0.92(0.90,0.95) | | |
| **Diabetes** | | | **0.026** |
| Borderline | 1.04(0.97,1.10) | | |
| No | 0.94(0.93,0.96) | | |
| Yes | 0.94(0.91,1.01) | | |
| **Hypertension** | | | 0.131 |
| No | 0.96(0.94,0.98) | | |
| Yes | 0.94(0.92,0.95) | | |
| **Hyperlipidemia** | | | 0.769 |
| No | 0.94(0.92,0.96) | | |
| Yes | 0.95(0.93,0.96) | | |
| | | 0.90  0.95  1.00  1.05  1.10 | |

**Fig 3. Subgroups analyses for the association between CDAI and anemia.** BMI, body mass index; PIR, poverty income ratio.

metabolic syndrome [12]. Similar, Ma et al in their study on the correlation between CDAI and coronary heart disease, they reported negative correlation, indicating that CDAI is inversely related to the incidence of coronary heart disease (Q4 vs Q1, OR: 0.65, 95%CI: 0.51–0.82, $P < 0.001$) [33]. While research on the relationship between CDAI and anemia is currently limited.

Our study reinforces these findings by showing an inverse relationship between CDAI levels and anemia in American adults (OR: 0.97, 95%CI: 0.96–0.98). Specifically, higher CDAI scores were linked to lower prevalence of anemia. Our findings align with those of Zhang et al, who reported that among 5,880 participants, higher CDAI was associated with reduced likelihood of renal anemia (adjusted OR: 0.96, 95% CI: 0.94–0.98) [34]. In addition, studies focusing on specific anemic populations, such as patients with β- thalassemia, provide further insight into the relationship between anemia severity and antioxidant defenses. Allen et al suggest that the severity of anemia in these patients is linked to depletion of antioxidant defenses, indicating that antioxidant supplementation may be beneficial [35]. Andrea et al summarize the intricate interactions between antioxidant-rich foods and gut microbiota, inflammation, and obesity, positing that plant-based antioxidant foods, such as vegetables, fruits, and nuts, are essential for health maintenance [36]. Meanwhile, Jacques et al investigated the impact of various nutrients, including antioxidants like vitamin A and vitamin C, on iron metabolism to mitigate the risk of anemia [37]. This finding aligns with our results, which show that higher

CDAI scores are associated with lower prevalence of anemia, thereby emphasizing the potential therapeutic role of antioxidants. Therefore, a balanced diet, such as the Mediterranean diet, along with specific foods like fish, fresh vegetables, and fruits, is recommended for individuals with low CDAI scores [36]. These foods are rich in essential components, including fiber, minerals, vitamins, and antioxidants, which can help prevent the occurrence of anemia.

Subsequently, this study explored the correlation between antioxidant components and anemia, discovering that vitamins E, zinc, carotene, and selenium were independently linked to anemia after adjusting for all potential confounders. Vitamin E is recognized as a crucial lipophilic antioxidant in biological membranes, capable of scavenging free radicals and acting as a chain-breaking antioxidant [38]. Severe vitamin E deficiency can lead to impaired immune response and hemolytic anemia caused by free radicals [39]. Sue et al highlighted that the lack of multiple trace element biomarkers (iron, zinc, selenium) was positively associated with anemia [40]. While Lisa et al showed that low zinc is an independent risk factor for anemia in school-age children and mediates the effect of low selenium on hemoglobin levels [41]. Carotene exert antioxidant effects through direct interactions with free radicals, which occur via electron or hydrogen atom transfer [42]. Additionally, carotene can be converted into vitamin A within the body, promoting iron absorption and metabolism, thereby exerting preventive effect against anemia [43]. However, these studies focused solely on specific antioxidants and did not examine potential synergies among various antioxidant nutrients. Given that our dietary intake typically includes multiple antioxidants, the CDAI seems to exert a comprehensive effect on individuals' pro-antioxidant status and underscores the advantages of thorough assessments of antioxidant exposure [17].

This study has several strengths. Firstly, it is the first cross-sectional survey to explore the relationship between CDAI and anemia, emphasizing the potential impact of dietary antioxidants in reducing anemia prevalence. Secondly, the findings indicate that the consumption of antioxidant- rich foods could potentially decrease the risk of anemia and offer dietary guidance to boost antioxidant intake for individuals with anemia.

However, this study also presents several limitations. Firstly, the dietary data in NHANES is based on self-reporting, which may introduce recall bias, and such errors are unavoidable. Secondly, the cross-sectional design complicates the establishment of causal relationship between CDAI and anemia, indicating that prospective multicenter studies will be necessary in the near future to validate our findings. Thirdly, the anemia data utilized in this study were derived from laboratory diagnoses and may not have accounted for the potential contributions of a history of anemia or biological differences, such as smoking or an increased incidence of thalassemia. Lastly, although this study accounted for numerous clinical variables, potential unmeasured confounders, such as medication use and laboratory indicators, including liver and kidney function, iron levels, and vitamins, may still influence CDAI levels and their correlation with outcomes. Consequently, future research should aim to explore the specific associations between these variables more comprehensively.

## Conclusion

This study revealed an inverse relationship between CDAI and the prevalence of anemia, with more pronounced effect observed in male nonsmokers and nondiabetic individuals. These findings offer valuable insights for healthcare providers to develop more targeted anemia screening and prevention strategies. Furthermore, Our study suggest the potential role of dietary modifications, particularly for individuals with lower CDAI scores, including the adoption of antioxidant-rich dietary patterns, such as iron supplements and plant-based diets, to help

reduce the prevalence of anemia. Additionally, further large-scale prospective studies are necessary to confirm these findings.

## Supporting information

**S1 Data. Detailed description of code used.**
(TXT)

**S1 Raw data. Original data for this study.**
(XLSX)

**S1 Table. The weighted logistic regression analysis of the association between CDAI and HB.**
(DOCX)

## Author Contributions

**Data curation:** Hui Xu.

**Formal analysis:** Qian Wu, Hui Xu.

**Investigation:** Jieyu Xia, Hui Xu, Guangyong Feng.

**Methodology:** Qian Wu, Jieyu Xia, Guangyong Feng.

**Project administration:** Xiaoxia Gou.

**Resources:** Zhiyu Wang, Guangyong Feng, Xiaoxia Gou.

**Software:** Zhiyu Wang, Hui Xu.

**Supervision:** Gang Huang.

**Validation:** Jieyu Xia, Gang Huang.

**Visualization:** Gang Huang.

**Writing – original draft:** Qian Wu, Zhiyu Wang, Xiaoxia Gou.

**Writing – review & editing:** Qian Wu, Xiaoxia Gou.

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
