## [Decision Letter · Decision Letter 0]

10 Oct 2024

PONE-D-24-28259

Association between composite dietary antioxidant indices and anemia: NHANES 2003–2018

PLOS ONE

Dear Dr. Wu,

Thank you for submitting your manuscript to PLOS ONE. After careful consideration, we feel that it has merit but does not fully meet PLOS ONE’s publication criteria as it currently stands. Therefore, we invite you to submit a revised version of the manuscript that addresses the points raised during the review process.

The manuscript has been evaluated by three reviewers, and their comments are available below.

The reviewers have raised a number of concerns that need attention. They request additional information on methodological aspects of the study (such as the anemia cut-off level and exclusion criteria), and raise concerns over the effect of recall bias.

Could you please revise the manuscript to carefully address the concerns raised?

We look forward to receiving your revised manuscript.

Kind regards,

Helen Howard

Staff Editor

PLOS ONE

“1.Guizhou Provincial Department of Science and Technology project

2.Science and Technology Program of Guizhou Anti-Cancer Association”

6. Please include your tables as part of your main manuscript and remove the individual files. Please note that supplementary tables (should remain/ be uploaded) as separate "supporting information" files

Reviewers' comments:

Reviewer's Responses to Questions

**Comments to the Author**

1. Is the manuscript technically sound, and do the data support the conclusions?

Reviewer #1: Yes

Reviewer #2: Yes

Reviewer #3: Partly

2. Has the statistical analysis been performed appropriately and rigorously? 

Reviewer #1: Yes

Reviewer #2: Yes

Reviewer #3: I Don't Know

3. Have the authors made all data underlying the findings in their manuscript fully available?

Reviewer #1: Yes

Reviewer #2: Yes

Reviewer #3: Yes

4. Is the manuscript presented in an intelligible fashion and written in standard English?

Reviewer #1: Yes

Reviewer #2: Yes

Reviewer #3: Yes

5. Review Comments to the Author

Reviewer #1: This manuscript provides a background on the topic, however, it need a clearer or specific research gab. Additionally, the authors should provide more details on the rationale for selecting the Composite Dietary Antioxidant Index (CDAI). About The CDAI, how to use CDAI, and if you compare it with another method, what kind of comparative advantage that you get.

Based on that survey, what kind of food source that lead to increasing level of six nutrients by considering age, sex, race/ethnicity, education level, body mass index (BMI), poverty income ratio (PIR),

Please write diet recommendation that is more effective and that is considered as a strategy in anemia prevention (add in your discuss)

Reviewer #2: 1. How can the 24-hour dietary recall interview method explain the diversity of daily consumption, how to control consumption on holidays, during parties, when dieting or fasting?

2. How to overcome recall bias in this study?

3. How to control that the results of this analysis do not occur by chance, not because of the large number of samples? Are there other dominant factors that can interfere with the relationship between the two, besides the existing variables?

4. How to improve the results of the relationship between the two, considering that the study design used was cross-sectional?

Reviewer #3: Dear Authors

I appreciate your efforts and hard work.

I have a few recommendations.

Rephrase the opening sentence of the abstract: "Recent studies have revealed that..." This statement requires references. You may report it as there is a documented association between,,,,

Add extra details to the method part.

Background

Please elaborate on the relationship between anemia and antioxidants, and how antioxidants diminish anemia. Protein synthesis? What biological reaction affected by the oxidative stress? What physiological pathway is affecte?

Methodology

i Have concern about the anemia cut score according to smoking status and living in high altitude areas? both affect the Hb cut score?

Also you studies only iron deficiency anemia? not all types? clarify and justify why?

In your exclusion criteria, other than excluding the missing data, do you include chronic GI diseases, malabsorption disorders, and pregnancy?

The disucsison

It would be beneficial to include earlier research on the relationship between diet consumption and anemia (which are numerous). To support your hypothesis, compare your results with those of these studies.

6. PLOS authors have the option to publish the peer review history of their article (what does this mean?). If published, this will include your full peer review and any attached files.

Reviewer #1: **Yes: **Betty Yosephin Simanjuntak

Reviewer #2: No

Reviewer #3: **Yes: **Manal Badrasawi

---

## [Author Response · Author response to Decision Letter 0]

13 Nov 2024

Dear Editors and Reviewers,

Thank you for giving us an opportunity to revise our manuscript. On behalf of all the contributing authors, I would like to express our sincere appreciations of your letter and reviewers’ constructive comments concerning our article entitled “Association between composite dietary antioxidant indices and anemia: NHANES 2003–2018” (Number: PONE-D-24-28259). Those comments are all valuable and very helpful for revising and improving our paper. We have studied comments carefully and have made extensive modifications which we hope meet with approval. The editors and reviewers’ comments are laid out below in blue font and specific concerns have been numbered. Our responses are provided in normal font, and the specific locations of changes and additions to the manuscript are highlighted in red text. Additionally, we have included grant numbers and funding sources in the funding information section. We have also uploaded the proof of polishing the manuscript. If there are any specific location problems, we will continue to modify. 

Thank you and best regards.

Yours sincerely,

Xiaoxia Gou, MD

Department of Head and Neck Oncology

The Second Affiliated Hospital of Zunyi Medical University

Zunyi, Guizhou 563000,China. 

E-mail: gouxx2020@163.com

Response to comments from Editors and Reviewers

Editorial comment

1.Comment: Please ensure that your manuscript meets PLOS ONE's style requirements, including those for file naming. The PLOS ONE style templates can be found at https://journals.plos.org/plosone/s/file?id=wjVg/PLOSOne_formatting_sample_main_body.pdf and https://journals.plos.org/plosone/s/file?id=ba62/PLOSOne_formatting_sample_title_authors_affiliations.pdf.

1.Response: Thank you for your kind reminder. In response to your suggestions, we revisited the PLOS ONE submission guidelines and implemented several changes, including partial rearrangements of text fonts, figure captions, references, and other elements within the manuscript.

2.Comment: Please ensure that you include a title page within your main document. Wedo appreciate that you have a title page document uploaded as a separate file,however, as per our author guidelines (http://journals.plos.org/plosone/s/submission-guidelines#loc-title-page) we do require this to be part of the manuscript file itself and not uploaded separately. Could you therefore please include the title page into the beginning of your manuscript file itself, listing all authors and affiliations.

2.Response: Thanks for your reminding. We will ensure that title page is included at the beginning of the manuscript.

3.Comment: We note that the grant information you provided in the ‘Funding Information’ and ‘Financial Disclosure’ sections do not match. When you resubmit, please ensure that you provide the correct grant numbers for the awards you received for your study in the ‘Funding Information’ section.

3.Response: We sincerely thank you for your valuable reminding and have added the grant number to the grant information section.

This study was supported by the National Natural Science Foundation of China (82460465); The Natural Science Foundation of Guizhou Province (Qian Ke He Basic Project ZK [2024]346); The Guizhou Anti-Cancer Association Science and Technology Plan Project (Anti-Cancer Association Science and Technology 006[2023]); The Natural Science Foundation of Guizhou Province (Qian Ke He Basic Project ZK [2023]525).

4.Comment: Thank you for stating the following financial disclosure: “1.Guizhou Provincial Department of Science and Technology project 2.Science and Technology Program of Guizhou Anti-Cancer Association” Please state what role the funders took in the study.  If the funders had no role, please state: "The funders had no role in study design, data collection and analysis, decision to publish, or preparation of the manuscript." If this statement is not correct you must amend it as needed. Please include this amended Role of Funder statement in your cover letter; we will change the online submission form on your behalf.

4.Response: Thank you for your kind reminder. The funder participated in this study, and the specific funding information has been matched and updated in the cover letter.

Eg: Xiaoxia Gou: The National Natural Science Foundation of China (82460465); The Natural Science Foundation of Guizhou Province (Qian Ke He Basic Project ZK [2024]346); The Guizhou Anti-Cancer Association Science and Technology Plan Project (Anti-Cancer Association Science and Technology 006[2023]); 

Gang Huang: The Natural Science Foundation of Guizhou Province (Qian Ke He Basic Project ZK [2023]525).

5.Comment: Please include a separate caption for each figure in your manuscript.

5.Response: Thanks for your reminding. Each figure in our manuscript is accompanied by distinct figure caption.

6.Comment: Please include your tables as part of your main manuscript and remove the individual files. Please note that supplementary tables (should remain/ be uploaded) as separate "supporting information" files

6.Response: Thank you once again for your kind reminder. We have included the tables as part of the manuscript and ensured that the supplementary tables are uploaded as separate ‘Supporting Information’ files.

Review Comments

Reviewer #1:

1.Comment: This manuscript provides a background on the topic, however, it need a clearer or specific research gab. Additionally, the authors should provide more details on the rationale for selecting the Composite Dietary Antioxidant Index (CDAI). About The CDAI, how to use CDAI, and if you compare it with another method, what kind of comparative advantage that you get.

1.Response: Thank you immensely for your inquiry, your insights are profoundly valuable. Indeed, our introduction section had overlooked critical aspects, particularly a detailed exploration of the CDAI and focus on comparative research. To clearly articulate the research gap and align with your recommendations, we have enriched our introduction section with the following revised content.

1)Based on your recommendations, we have added a section in the Introduction section about specific research gab, with the specifics outlined below: 

① (Manuscript, Introduction section, page 3, line 69-72): The prevalence and incidence of anemia have risen significantly, attributed to combination of increased nutrient deficiencies, chronic diseases, inherited hemoglobin disorders, and the use of specific medications [4,5].

Reference:

[4]. Chen S, Xiao J, Cai W, Lu X, Liu C, Dong Y, et al. Association of the systemic immune-inflammation index with anemia: a population-based study. Frontiers in immunology. 2024;15:1391573. http://dx.doi.org/10.3389/fimmu.2024.1391573 PMID: 38799419.

[5]. Stevens GA, Paciorek CJ, Flores-Urrutia MC, Borghi E, Namaste S, Wirth JP, et al. National, regional, and global estimates of anaemia by severity in women and children for 2000-19: a pooled analysis of population-representative data. The Lancet Global health. 2022;10(5):e627-e39. http://dx.doi.org/10.1016/s2214-109x(22)00084-5 PMID: 35427520.

② (Manuscript, Introduction section, page 3, line 75-78): Early recognition of anemia presents an opportunity to delay or prevent the onset of the disease and enhance treatment outcomes. Consequently, identifying new indicators closely associated with anemia is of great significance for developing more effective anemia prevention strategies [4].

Reference:

[4]. Chen S, Xiao J, Cai W, Lu X, Liu C, Dong Y, et al. Association of the systemic immune-inflammation index with anemia: a population-based study. Frontiers in immunology. 2024;15:1391573. http://dx.doi.org/10.3389/fimmu.2024.1391573 PMID: 38799419.

③(Manuscript, Introduction section, page 4, line 91-93): As research enhances our understanding of nutrition and oxidative stress, there is an increasing interest in the role of antioxidant-rich diets in the prevention of anemia.

2) We revised the paper and added necessary descriptions about more details for the CDAI in the introduction section and discussion section. The details are as follows :

① (Manuscript, Introduction section, page 4, line 94-96): The composite dietary antioxidant index (CDAI) was developed by Wright et al as a tool to assess antioxidant intake, serving as a composite score to evaluate dietary antioxidant consumption [16].

Reference:

[16] Wright ME, Mayne ST, Stolzenberg-Solomon RZ, Li Z, Pietinen P, Taylor PR, et al. Development of a comprehensive dietary antioxidant index and application to lung cancer risk in a cohort of male smokers. American journal of epidemiology. 2004;160(1):68-76. http://dx.doi.org/10.1093/aje/kwh173 PMID: 15229119. 

② (Manuscript, Introduction section, page 4, line 98-106): Wang et al posited that the CDAI was positively correlated with lower prevalence of chronic kidney disease among American adults [17]. Similarly, Yu et al discovered that elevated CDAI scores were linked to decreased risk of colorectal cancer (CRC) and concluded that food-based antioxidants might contribute to lowering the risk of CRC in the general population [18]. Additionally, another study indicated that higher intake of dietary antioxidants, assessed through the Dietary Antioxidant Quality Score (DAQS) and CDAI, was associated with reduced risk of all-cause and cardiovascular disease mortality in adults with diabetes [19]. 

Reference:

[17]. Wang M, Huang ZH, Zhu YH, He P, Fan QL. Association between the composite dietary antioxidant index and chronic kidney disease: evidence from NHANES 2011-2018. Food & function. 2023;14(20):9279-86. http://dx.doi.org/10.1039/d3fo01157g PMID: 37772927.

[18]. Yu YC, Paragomi P, Wang R, Jin A, Schoen RE, Sheng LT, et al. Composite dietary antioxidant index and the risk of colorectal cancer: Findings from the Singapore Chinese Health Study. International journal of cancer. 2022;150(10):1599-608. http://dx.doi.org/10.1002/ijc.33925 PMID: 35001362.

[19]. Wang W, Wang X, Cao S, Duan Y, Xu C, Gan D, et al. Dietary Antioxidant Indices in Relation to All-Cause and Cause-Specific Mortality Among Adults With Diabetes: A Prospective Cohort Study. Front Nutr. 2022;9:849727. http://dx.doi.org/10.3389/fnut.2022.849727 PMID: 35600816.

③ (Manuscript, Discussion section on Page 9, line 251-256): In this study, we focus on the CDAI for two primary reasons. Firstly, CDAI serves as a crucial indicator of dietary antioxidant capacity. Unlike single dietary antioxidant indicators, CDAI encompasses key antioxidant nutrients, including vitamin A, vitamin C, vitamin E, zinc, selenium, and carotene, thereby facilitating more comprehensive evaluation of overall diet quality. Secondly, research in related fields has garnered significant attention. 

④ (Manuscript, Discussion section on Page 9, line 259-262): Similar, Ma et al in their study on the correlation between CDAI and coronary heart disease, they reported negative correlation, indicating that CDAI is inversely related to the incidence of coronary heart disease (Q4 vs Q1, OR: 0.65, 95%CI: 0.51 - 0.82, P < 0.001) [33]. 

Reference:

[33]. Ma R, Zhou X, Zhang G, Wu H, Lu Y, Liu F, et al. Association between composite dietary antioxidant index and coronary heart disease among US adults: a cross-sectional analysis. BMC public health. 2023;23(1):2426. http://dx.doi.org/10.1186/s12889-023-17373-1 PMID: 38053099.

⑤ (Manuscript, Discussion section, page 10, Line 264-269) : Our study reinforces these findings by showing an inverse relationship between CDAI levels and anemia in American adults (OR: 0.97, 95%CI: 0.96 - 0.98). Specifically, higher CDAI scores were linked to lower prevalence of anemia. Our findings align with those of Zhang et al, who reported that among 5,880 participants, higher CDAI was associated with reduced likelihood of renal anemia (adjusted OR: 0.96, 95% CI: 0.94 - 0.98) [34]. 

Reference:

[34]. Zhang L, Yin D, Zhu T, Geng L, Gan L, Ou S, et al. Composite dietary antioxidant index is associated with renal anemia: a cross-sectional study. International urology and nephrology. 2024. http://dx.doi.org/10.1007/s11255-024-04157-8 PMID: 39044023.

⑥ (Manuscript, Discussion section on Page 10, line 279-281): This finding aligns with our results, which show that higher CDAI scores are associated with lower prevalence of anemia, thereby emphasizing the potential therapeutic role of antioxidants. 

⑦ (Manuscript, Discussion section on Page 11, line 299-304): However, these studies focused solely on specific antioxidants and did not examine potential synergies among various antioxidant nutrients. Given that our dietary intake typically includes multiple antioxidants, the CDAI seems to exert a comprehensive effect on individuals’ pro-antioxidant status and underscores the advantages of thorough assessments of antioxidant exposure [17].

Reference:

[17]. Wang M, Huang ZH, Zhu YH, He P, Fan QL. Association between the composite dietary antioxidant index and chronic kidney disease: evidence from NHANES 2011-2018. Food & function. 2023;14(20):9279-86. http://dx.doi.org/10.1039/d3fo01157g PMID: 37772927.

2.Comment: Based on that survey, what kind of food source that lead to increasing level of six nutrients by considering age, sex, race/ethnicity, education level, body mass index (BMI), poverty income ratio (PIR), Please write diet recommendation that is more effective and that is considered as a strategy in anemia prevention (add in your discuss). 

2.Response: Based on your recommendations, we have checked the literature carefully and have revised the discussion to include specific dietary recommendations aimed at anemia prevention.

①(Manuscript, Discussion section, page 10, line 273-276): Andrea et al summarize the intricate interactions between antioxidant-rich foods and gut microbiota, inflammation, and obesity, positing that plant-based antioxidant foods, such as vegetables, fruits, and nuts, are essential for health maintenance [36]. 

② (Manuscript, Discussion section, page 10, line 281-285): Therefore, a balanced diet, such as the Mediterranean diet, along with specific foods like fish, fresh vegetables, and fruits, is recommended for individuals with low CDAI scores [36]. These foods are rich in essential components, including fiber, minerals, vitamins, and antioxidants, which can help prevent the occurrence of anemia. 

Reference:

[36]. Deledda A, Annunziata G, Tenore GC, Palmas V, Manzin A, Velluzzi F. Diet-Derived Antioxidants and Their Role in Inflammation, Obesity and Gut Microbiota Modulation. Antioxidants (Basel, Switzerland). 2021;10(5). http://dx.doi.org/10.3390/antiox10050708 PMID: 33946864.

③ We have included effective dietary recommendations for the prevention of anemia in the conclusion section. Please refer to the conclusion section, page 12, line 325-331: This study revealed an inverse relationship between CDAI and the prevalence of anemia, with more pronounced effect observed in male nonsmokers and nondiabetic individuals. These findings offer valuable insights for healthcare providers to develop more targeted anemia screening and prevention strategies. Furthermore, Our study suggest the potential role of dietary modifications, particularly for individuals with lower CDAI scores, including the adoption of antioxidant-rich dietary patterns, such as iron supplements and plant-based diets, to help reduce the prevalence of anemia. 

Reviewer #2:

1.Comment: How can the 24-hour dietary recall interview method explain the diversity of daily consumption, how to control consumption on holidays, during parties, when dieting or fasting? 

1. Response: Thank you immensely for your inquiry, your insights are profoundly valuable.

1) In line with previous research, participants completed two such interviews. Please refer Manuscript, Method section, page 5, line 137-145: Data on dietary antioxidant intake were obtained from the average of two 24-hour dietary recall interviews conducted as part of the NHANES. The initial dietary recall was performed at mobile examination center (MEC) by trained interviewers who adhered to standardized protocol. During this face-to-face interview, detailed information regarding all food and beverages consumed by participants over the past 24 hours was collected

---

## [Decision Letter · Decision Letter 1]

10 Dec 2024

Association between composite dietary antioxidant indices and anemia: NHANES 2003–2018

PONE-D-24-28259R1

Dear Dr. Wu,

We’re pleased to inform you that your manuscript has been judged scientifically suitable for publication and will be formally accepted for publication once it meets all outstanding technical requirements.

Kind regards,

Mehmet Baysal

Academic Editor

PLOS ONE

Additional Editor Comments (optional):

Reviewers' comments:

Reviewer's Responses to Questions

**Comments to the Author**

1. If the authors have adequately addressed your comments raised in a previous round of review and you feel that this manuscript is now acceptable for publication, you may indicate that here to bypass the “Comments to the Author” section, enter your conflict of interest statement in the “Confidential to Editor” section, and submit your "Accept" recommendation.

Reviewer #1: All comments have been addressed

Reviewer #3: All comments have been addressed

2. Is the manuscript technically sound, and do the data support the conclusions?

Reviewer #1: Yes

Reviewer #3: Yes

3. Has the statistical analysis been performed appropriately and rigorously? 

Reviewer #1: Yes

Reviewer #3: Yes

4. Have the authors made all data underlying the findings in their manuscript fully available?

Reviewer #1: Yes

Reviewer #3: Yes

5. Is the manuscript presented in an intelligible fashion and written in standard English?

Reviewer #1: Yes

Reviewer #3: Yes

6. Review Comments to the Author

Reviewer #1: The authors have enriched detail and clearly in all sections that I comment. This manuscript can accepted to publish

Reviewer #3: No more comments, the authores replied to the comments suffeciently. I think the decision may be made according to the journal standards and scope

7. PLOS authors have the option to publish the peer review history of their article (what does this mean?). If published, this will include your full peer review and any attached files.

Reviewer #1: **Yes: **Betty Yosephin Simanjuntak

Reviewer #3: No

---

## [Editor Report · Acceptance letter]

15 Dec 2024

PONE-D-24-28259R1 

PLOS ONE

Dear Dr. Wu, 

I'm pleased to inform you that your manuscript has been deemed suitable for publication in PLOS ONE. Congratulations! Your manuscript is now being handed over to our production team.

Kind regards, 

on behalf of

Dr. Mehmet Baysal 

Academic Editor

PLOS ONE